# Antibacterial Activity from *Momordica charantia* L. Leaves and Flavones Enriched Phase

**DOI:** 10.3390/pharmaceutics14091796

**Published:** 2022-08-26

**Authors:** Abraão de Jesus B. Muribeca, Paulo Wender P. Gomes, Steven Souza Paes, Ana Paula Alves da Costa, Paulo Weslem Portal Gomes, Jéssica de Souza Viana, José Diogo E. Reis, Sônia das Graças Santa R. Pamplona, Consuelo Silva, Anelize Bauermeister, Lourivaldo da Silva Santos, Milton Nascimento da Silva

**Affiliations:** 1Institute of Exact and Natural Sciences, Federal University of Pará, Augusto Corrêa, 01, Guamá, Belém 66075-110, PA, Brazil; 2Collaborative Mass Spectrometry Innovation Center, University of California San Diego, La Jolla, San Diego, CA 92093, USA; 3Skaggs School of Pharmacy and Pharmaceutical Sciences, University of California, La Jolla, San Diego, CA 92093, USA; 4Department of Natural Science, Campus XIX, State University of Pará, Rodovia PA 154, Km 28, Cajú, Salvaterra 66860-000, PA, Brazil; 5Institute of Biology, University of Campinas, Monteiro Lobato, 255, Barão Geraldo, Campinas 13083-862, SP, Brazil; 6Pharmaceutical Science Post-Graduation Program, Faculty of Pharmacy, Federal University of Pará, Belém 66075-110, PA, Brazil; 7Institute of Biomedical Sciences, University of São Paulo, São Paulo 05508-900, SP, Brazil

**Keywords:** glycosylated flavones, molecular networking, dereplication, antibacterial activity

## Abstract

*Momordica charantia* L. (Cucurbitaceae) is a plant known in Brazil as “melão de São Caetano”, which has been related to many therapeutic applications in folk medicine. Herein, we describe antibacterial activities and related metabolites for an extract and fractions obtained from the leaves of that species. An ethanolic extract and its three fractions were used to perform in vitro antibacterial assays. In addition, liquid chromatography coupled to mass spectrometry and the molecular networking approach were used for the metabolite annotation process. Overall, 25 compounds were annotated in the ethanolic extract from *M. charantia* leaves, including flavones, terpenes, organic acids, and inositol pyrophosphate derivatives. The ethanolic extract exhibited low activity against *Proteus mirabilis* (MIC 312.5 µg·mL^−1^) and *Klebsiella pneumoniae* (MIC 625 µg·mL^−1^). The ethyl acetate phase showed interesting antibacterial activity (MIC 156.2 µg·mL^−1^) against *Klebsiella pneumoniae*, and it was well justified by the high content of glycosylated flavones. Therefore, based on the ethyl acetate phase antibacterial result, we suggest that *M. charantia* leaves could be considered as an alternative antibacterial source against *K. pneumoniae* and can serve as a pillar for future studies as well as pharmacological application against the bacteria.

## 1. Introduction

Clinical infections caused by resistant bacteria have become a major public health concern worldwide, and approximately 700,000 deaths per year are caused by this type of bacteria [1]. It is estimated that by 2050, there will be more than 10 million deaths/year credited to ‘superbugs’, with the expectation of the highest rate being reported in developing countries [2]. The demand for new antimicrobial agents has been growing at the same time as the discovery and advancement of multi-resistant bacteria. Therefore, researchers have been focusing a great effort on the search for new therapeutic alternatives against multidrug-resistant bacteria [3]. In this sense, plant derivatives appear as potential sources of new drugs that act in different ways to deactivate or block the growth of such pathogens [4].

The Cucurbitaceae family includes several species widely distributed throughout the tropical and subtropical regions. There are antimicrobial activities associated with crude extracts and isolated metabolites of species from this family [5]. Among the species, *Momordica charantia* L., popularly known as “melão de São Caetano”, is considered an invasive plant in Brazil and can be found in different places in the country. It is frequently grown in orchards and coffee plantations, or even on fences and debris in abandoned land [6], and is composed of compound classes such as terpenoids, saponins, and flavonoids [5,7]. This species can be highlighted due to its antimicrobial [5], nutraceutical and inflammatory properties [8], as well as healing of gastric ulcers [9], rheumatism [10], and so on. In developing countries, *M. charantia* has been employed in folk medicine for several other pharmacological purposes, such as to treat toothache, diarrhea, furuncle, cancer, hypertension, obesity, bacterial and viral infections, diabetes, pneumonia, and even AIDS [11,12,13,14,15,16].

Although the chemical characterization of such plants can lead to new pharmaceuticals, it remains challenging due to the presence of several components with different physical – chemical properties, including many in relatively small quantities [17]. Nuclear magnetic resonance (NMR) techniques provide valuable structural information used for structure characterization and therefore, provide unequivocal identification. Liquid chromatography coupled to mass spectrometry (LC-MS) plays a crucial role as it is compatible with chromatographic techniques that allow the separation of compounds in the sample, leading to a deeper investigation of the whole chemical content due to the high sensitivity of MS that can detect metabolites of picomole to femtomole levels in some cases [18]. Furthermore, MS allows the detection of metabolites as ions in the form of mass to charge ratio (*m*/*z*) [19], and also allows the fragmentation of the ions at the gas phase by collision-induced dissociation (CID), providing MS/MS spectra that contain valuable information to contribute to annotation of the chemical structure [20]. It is worth mentioning that the acquisition of scan spectra by mass spectrometry is fast, and a single chromatographic run can provide thousands of MS spectra. Therefore, methods that allow a fast and easy analysis of such data are welcome in this field.

In this context, molecular networking, a tool from the Global Natural Products Social Molecular Networking (GNPS) [21] infrastructure, allows the visualization of MS/MS data by organizing it by spectral similarity. This is a very effective strategy, especially because establishing a similarity relationship provides organization into molecular families (usually related chemical structures), and, therefore, finds related metabolites in the dataset even in small amounts. GNPS also contains a spectral library of known compounds, and the molecular networking allows automatic searches in the spectral library, which contributes to speed up the dereplication of known compounds, a phytochemical approach that has been widely used in recent decades, including the modest contribution of these authors [22,23,24,25,26], and which is also unknown by recognition of the analogs into the molecular families.

Based on the antimicrobial potential already reported for *M. charantia*, this study describes the evaluation of in vitro antibacterial activity for the ethanolic extract and fractions from the leaves of *M. charantia*. In addition, liquid chromatography coupled to high-resolution mass spectrometry (LC-HRMS) analyses allowed the chemical characterization of the samples.

## 2. Materials and Methods

### 2.1. Botanical Material and Extraction

Leaves of *M. charantia* were collected in the municipality of Soure (0°23′50′′ S and 49°27′02′′ W), Marajó Island, PA, Brazil. The botanical material was incorporated (voucher: MFS009218) in the Prof. Dr. Marlene Freitas da Silva (MFS) Herbarium of the State University of Pará. Permission to access the Brazilian genetic patrimony was provided by SISGEN (A695619).

The leaves were washed with water from the Direct-Q5 system (Millipore, Darmstadt, Germany) and decontaminated with sodium hypochlorite solution (NaOCl, 0.1%) acquired from Dinâmica (Jaraguá do Sul, SC, Brazil). The samples were dried in an air circulation oven (Quimis, Brazil) at 45 °C until constant weight. The dry material was crushed in ball mills up to a granulometry of 60 to 100 μm, obtaining 146.54 g. The mass was subjected to extraction with 1 L of ethanol (Tedia, Fairfield, OH, USA) at room temperature for 24 h (2×), and 35.30 g of crude extract was obtained after the solvent evaporation process. Thus, 1 g of ethanolic extract (EE) was mixed with a hydroalcoholic solution, consisting of 60 mL of ultra-pure water, 20 mL of ethanol, and 1 mL of hydrochloric acid (Dinâmica, Indaiatuba, SP, Brazil). The resulting solution was subjected to liquid – liquid partition (LLP) to obtain the hexane (PhHex), ethyl acetate (PhEA), and hydroalcoholic (PhWOH) phases, respectively.

### 2.2. Liquid Chromatography-High Resolution Mass Spectrometry Analysis

The analyses were performed on a Xevo G2-S QqTOF mass spectrometer (Waters Corp., Milford, MA, USA) equipped with a LockSpray source. The instrument was calibrated with a mass of reference (leucine-enkephalin) utilized for accurate mass measurements. MassLynx 4.1 software was used for system control and data acquisition. The samples were analyzed in a BEH C18 column (Waters Corp.; 50 mm; 2.1 mm; 1.7 μm particle size) using ultra-pure water (solvent A) and acetonitrile (solvent B), both containing 0.1% formic acid. The column temperature was maintained at 40 °C. Linear gradient elution was performed with a flow of 300 μL/min and 5 – 95% of solvent B in 20 min. The injection volume of the samples was 5 μL. The mass spectra data were recorded in a negative ionization mode (ESI) for a mass range from *m/z* 50 to 1200. The source temperature was set to 120 °C with a cone gas flow of 50 L/h. The desolvation gas flow was set to 600 L/h at a temperature of 150 °C. The capillary was set at 3.0 kV with cone voltage at 40 V. The settings of the data-dependent acquisition (DDA) experiments were: centroid format, number of ions selected 5 (Top5 experiment), the normalized collision energy (NCE) was set to 10, 20, 30, 40 and 50, scan rate of 0.5 sec, charge states of +1 and +2, tolerance window of ±0.2 Da and peak extract window of 2 Da, tolerance of deisotope ± 3 Da, extraction tolerance of deisotope 6 Da.

### 2.3. Mass Spectrum Data Treatment

The raw files of the EE and PhEA acquired in the Xevo G2-S QTof mass spectrometer (Waters Corp., Milford, MA, USA) were converted into mzML format using the software MS Convert of the ProteoWizard package [27] and were processed with the software MZmine 2.53 version [28]. The limit for the detection of ions in negative mode at the MS^1^ level was set at 1.0 × 10^3^ and MS^2^ at 5.0 × 10^1^. Chromatograms were constructed using ADAP with a minimum group number of 3 and a minimum group intensity limit of 1.0 × 10^3^, a min highest intensity of 3.0 × 10^3,^ and an *m*/*z* tolerance of 10.0 ppm. The local minimum search algorithm was used to deconvolve the chromatogram, with an *m*/*z* tolerance of 0.5 for the pairing of MS^2^ and 0.2 min for RT. Isotopes were detected using a peak window with a tolerance of 10.0 ppm, an RT tolerance of 0.5 min, a maximum charge of 1. For peak alignment, the tolerance of *m*/*z* 10 ppm was used, scores for *m*/*z* of 75 and 25 for RT with a tolerance of 0.2 min. The resulting list was filtered to remove duplicates and lines with no associated MS^2^ spectrum. Then, gap filling was used to fill in the gaps in the peak list. The resultant files were exported using FBMN-GNPS.

### 2.4. Molecular Networks

The molecular network was built from the mgf and CSV files exported from MZmine. We used metadata to organize metabolite information according to the online workflow (available online: https://ccms-ucsd.github.io/GNPSDocumentation/, accessed on 30 January 2022) available on the GNPS website (available online: http://gnps.ucsd.edu, accessed on 8 July 2022) [21]. The tolerance of *m/z* for the precursor ion was adjusted to 0.02 Da and for fragment ion to 0.02 Da. Minimum cosine score above 0.5 and the minimum number of fragment ions were fixed on 4. The spectra on the network were then searched in the GNPS spectral libraries. The database spectra were filtered with a minimum cosine score above 0.6 and a minimum of 4 fragment ions correspondence. The job is available online at the link: https://gnps.ucsd.edu/ProteoSAFe/status.jsp?task=4890e08934bb4334b8738a346c10e7c4, accessed on 8 July 2022. The results were visualized and organized using Cytoscape version 3.8.2 (Seattle, WA, USA) [29]. Lastly, the correlation between PhEA and bioactivity score of the metabolites was obtained from NPAnalyst [30].

### 2.5. In Vitro Antibacterial Assay

Three strains of bacteria were used to evaluate the potential of the extracts: *Staphylococcus aureus* ATCC 25923, and *Klebsiella pneumoniae* ATCC 700603, provided by Instituto Evandro Chagas, Pará State (Brazil) and the bacteria *Proteus mirabilis* LACEN 8/7 (human isolated) provided by the Central Laboratory of Pará State collection. These microorganisms were selected for being common pathogens that can infect humans, animals or plants. The pure cultures were maintained by routine sub-culturing at one-week intervals in BHI broth (Brain Heart Infusion, Kasvi, Spain), incubated at 37 °C, and spiked for 24 h for their metabolic activation. The minimum inhibitory concentration (MIC) and minimum bactericidal concentration (MBC) were conducted using a method approved by the Clinical and Laboratory Standards Institute in 96-well microtitration plates [31].

### 2.6. Determination of Minimum Inhibitory Concentration (MIC) and Minimum Bactericidal Concentration (MBC)

The antimicrobial susceptibility test was conducted using a method approved by the Clinical and Laboratory Standards Institute [31]. The tests were carried out with a stock solution of 5 mg·mL^−1^ of the crude extract/phases using the successive dilution method to obtain the concentrations from 2500 to 78.1 µg·mL^−1^. Ciprofloxacin (Medley, Brazil) and vancomycin (1 mg·mL^−1^ each) were used as positive controls and BHI (Brain Heart Infusion) culture medium was used as negative control.

A total of 5 mg of extract/phases was dissolved in 100 μL of DMSO (Neon, Brazil) contained in Eppendorf tubes. Then, 900 μL of sterile BHI was added and stirred for better homogenization. In a 96-well microtitration plate (KASVI, Brazil), 100 μL of BHI broth was added to each well. Then, 100 μL of solution containing the samples was added to the first well of each column.

In each well, 5 μL of the bacterial suspension (10^4^ Colony Forming Unit CFU/mL as required by CLSI) was inoculated and adjusted to 0.5 McFarland Standard scale and then the plates were incubated at 37 °C for 24 h. The results were read by adding 10 μL of TTC (2,3,5-triphenyltetrazolic chloride, NEON, Brazil). To prepare the dye, 0.2 g of TTC was added to the penicillin type flask containing 10 mL of sterile distilled water. The final solution showed a translucent color, and when in contact with environments where there are microorganisms presents a red color.

The type of activity presented in each concentration (bacteriostatic or bactericidal) was checked. In the cavities where there was no red color caused by the reaction of TTC with the bacteria, 5 μL of the volume contained in the wells was (re) inoculated in a Petri dish containing BHI agar culture medium and incubated at 37 °C for 24 h. Wells where bacteria grew indicated a bacteriostatic effect at that concentration and wells without bacteria indicated a bactericidal effect.

## 3. Results and Discussion

### 3.1. Antibacterial Activity

The EE and the PhEA of *M. charantia* showed antimicrobial activity and, therefore, they were selected to be analyzed by LC-MS/MS. Both positive controls of vancomycin and ciprofloxacin had a MIC of 7.8 µg·mL^−1^. The EE demonstrated a good bactericidal effect against *Klebsiella pneumoniae* and *Proteus mirabilis*. The PhEA stands out for presenting the best result with a MIC of 156.2 µg·mL^−1^ for the *K. pneumoniae*, suggesting that this presents bioactive substances, possibly the glycosylated flavones, responsible for such activity (Table 1). We highlighted the bactericidal effect of *Klebsiella Pneumonie,* and *Proteus Mirabilis*, both Gram-negative strains. The literature [32] reports that the cell wall of Gram-negative bacteria acts as a barrier to a number of substances, including antibiotics. However, recently it was confirmed that quercetin derivatives have strong antibacterial action against Gram-negative bacteria [33], and the ethyl acetate phase (PhEA, see Table 1) described in this study showed high content of quercetin (16) derivatives, i.e., quercetin-*O*-sambubioside (4), quercetin-*O*-glucoside (6), quercetin-*O*-glucosyl-6′′-acetate (9), and quercetin-O-acetylpentoside (13). MIC values < 100 µg·mL^−1^ are considered significant antimicrobials; moderate inhibitors present MIC in the range of 100 to 625 µg·mL^−1^; and inhibitors with MIC > 625 µg·mL^−1^ are considered weak [34]. In this sense, the ethyl acetate phase showed moderate activity (Table 1) against *K. pneumoniae* (MIC of 156.2 µg·mL^−1^), while the ethanolic extract showed moderate activity against *P. mirabilis* (312.5 µg·mL^−1^) and weak activity against *K. pneumoniae* and *S. aureus* (625 µg·mL^−1^). We emphasize that the MIC value against *K. pneumoniae* is in the range moderate–significant, which characterize PhEA, a source of candidate inhibitors of important hospital bacteria.

In previous reports, antimicrobial activity for leaves, fruits and seeds was reported against some clinically important bacteria [20]. The extract of the leaves showed the main results of being a potent inhibitor for *Staphylococcus aureus*, moderate for *Staphylococcus epidermidis* and weak for *Candida albicans*. Studies with extracts of the seed showed interesting activities against *Escherichia coli*, *Salmonella typhi*, and *Staphylococcus aureus*, but less activity against *Pseudomonas aeruginosa* [35].

### 3.2. Identification of Chemical Constituents

The total ion profiles of the EE and PhEA were recorded from 50 to 1200 Da in 20 min (Figure 1). The molecular formulas, main fragment ions, and putative names are shown in Table 2. A total of 32 major metabolites were detected in the EE. Of these, 25 compounds were annotated in level 2 and 3 of identification according to MSI [36] based on HRMS and MS/MS data and the literature; most of these compounds have well characterized fragmentation (MS/MS) profiles [37].

### 3.3. Molecular Networking (GNPS Annotation)

The molecular networking created with EE and PhEA showed 224 parent ions after removing the blank. Seven compounds (**6**, **7**, **8**, **10**, **12**, **23** and **32**) including isomers were annotated based on MS^2^ data available in the GNPS spectral libraries. A family of flavones was reported, and the structure of the compounds is shown in Figure 2.

The peak 6 [M − H]^−^ of *m*/*z* 463.0869 with the main fragments *m/z* 301, 271, and 179 was annotated as quercetin-*O*-glucoside. The loss of the sugar unit [(M − H) − H_2_O]^−^ explains the fragment of *m*/*z* 301. However, the position of the hydroxyl groups in ring B, as well as the glycosyl moiety, cannot be confirmed with only MS/MS data. The loss of H_2_O following CO [(M − H) − H_2_O − CO]^−^ justifies the *m/z* 271 [39]; lastly, the loss of C_7_H_6_O_2_ on the B ring by retrocyclisation [39] explains the fragment of *m/z* 179. The peak 7 [M − H]^−^ of *m/z* 579.1355 was annotated as kaempferol *O*-glucoside-*O*-pentoside (product ions: *m/z* 463, 399, 327, 285, 151, 109). The losses of C_5_H_8_O_3_ [(M − H) − C_5_H_8_O_3_]^−^ and C_7_H_16_O_5_ [(M − H) − C_5_H_8_O_3_]^−^ confirm the presence of two sugar units in the molecule, however, as discussed before, the positions of the sugar moieties cannot be certainly confirmed. Furthermore, characteristic fission from sugars (^0,2^ X_1_ mechanism) [40] suggests the *m/z* 327, the ion of *m/z* 285, occurs by losses of two sugar units, and the loss of C_8_H_6_O_2_ referred to the C ring [(M − H) − C_8_H_6_O_2_]^−^ explains the ion of *m*/*z* 151^.^ The base structure coumarin is identified by loss of C_9_H_4_O_4_ to the B ring [(M − H) − C_9_H_4_O_4_]^−^ characterizing the fragment of *m/z* 109. The peak 8 [M − H]^−^ of *m/z* 593.15 was annotated as luteolin-*O*-rutinoside and this molecule showed main fragments of *m/z* 547, 447, 357, 327, 285. The loss of C_2_H_6_O is very common in glycosylated flavones [41], which explains the fragment of *m/z* 547. In addition, from *m/z* 593 to 447 loss of C_6_H_10_O_4_ [(M − H) − 146]^−^ occurs and from *m/z* 447 to 285 loss of another sugar unit occurs. Lastly, *m*/*z* 285 is confirmed as the aglycone peak. The other fragments are very well discussed in the literature [40]; losses of C_3_H_6_O_3_ [(M – H) − 146 − 90]^−^ and CH_2_O [(M − H) − 146 − 90 − 30]^−^ suggested the ions of *m/z* 357 and 327, respectively. The peak 10 [(M − H)]^−^ of *m*/*z* 447.0926 was annotated as kaempferol-*O*-glucoside and the fragments of *m*/*z* 327, 284, 255, and 227 are very well discussed in the literature [39,42]. In summary, fission of kind ^0,2^ X_1_ occurs in the glycoside to the ion *m/z* 327, then, the aglycone (*m*/*z* 284) is confirmed by radical cleavage [^3^Y_0_ − H]^−^ followed by a loss of CH_2_O [^3^Y_0_ – CH_2_O]^−^ and CO [^3^Y_0_ − CH_2_O − CO]^−^ to the fragments of *m/z* 255 and 227. The peak 12 [M − H]^−^ of *m/z* 477.1035 was annotated as isorhamnetin-*O*-glucoside. The loss of C_2_H_6_O in the glucoside [41] explains the ion *m/z* 431 [(M − H) − 46]^−^ and the ^0,2^ X_1_ mechanism confirms the loss of C_4_H_8_O_4_ [(M − H) − 120]^−^ to the ion of *m/z* 357. The loss of glucoside occurs as *m/z* 315 [(M − H) − C_6_H_10_O_5_]^−^ followed by loss of radical CH_3_ from *m/z* 315 [(M − H) − C_6_H_10_O_5_ − CH_3_]^•−^. Aglycone corresponds to the ion of *m/z* 285; ion of *m/z* 271 was formed by a loss of CO_2_ [(M − H) − C_6_H_10_O_5_ − CO_2_]^−^ and retro-Diels-Alder (RDA) from *m/z* 151 [43,44].

The peak 32 [M − H]^−^ of *m/z* 571.2882 was characterized as 1-hexadecanoyl-sn-glycero-3-phospho-(1′-myo-inositol), an important inositol pyrophosphates derivative present in plants as a signaling metabolite [45,46,47]. While information about this new class of molecules in plants is still scarce, the enzymes responsible for their synthesis have recently been elucidated [48]. In this sense, 32 is being reported for the first time in the genus. Despite the report of some sulfur-derived compounds in the *M. charantia* sample [49], they were not observed in this work.

A molecular family of glycosylated terpene derivatives (Figure 3) was detected in the EE. The ion of *m*/*z* 385.1856 is referent from peak 3, identified as hydroxy-2,4,4-trimethyl-3-(3-oxobutyl)-2-cyclohexen-1-one glucoside, and the pathway of fragmentation has been shown in previous reports [50]. The fragmentation of the ion *m/z* 385 generated an ion at *m/z* 223, which refers to a neutral loss of a glycoside [(M − H) − 162]^−^. A loss of H_2_O indicates the ion *m/z* 205 followed by losses C_2_H_2_O and C_3_H_8_ suggesting the ions of *m/z* 163 [(M − H) − 42]^−^ and 119 [(M − H) − 42 − 44]^−^, respectively. Lastly, *m/z* 113 is explained by ring fission and loss of C_7_H_10_O [(M − H) − 110]^−^. This compound is included in the monoterpenes class and derivatives already reported in the literature from the Cucurbitaceae family [51].

According to Figure 2, the compounds **3**, **22**, **25**, and **27** showed the presence of glycoside. In this sense, following the chemosystematic from the genus, for example, the compound **25** was characterized as momordicoside L isomer, a cucurbitane-type triterpenoid already reported for the species [52]. According to the literature [52], the most intense product ion of *m/z* 471 corresponds to the aglycone after the loss of glycoside [(M − H) − Glc]^−^, and in addition a loss of C_3_H_6_O (propan-2-one) in C-5 characterized the ion of *m/z* 575 [(M − H) − 58]^−^ following loss of C_2_H_2_ to the fragment of *m*/*z* 549. Lastly, the ion of *m/z* 343 occurs by loss of C_8_H_16_O [(M − H) − Glc − 128]^−^ from the aglycone. Furthermore, this highlights that the cucurbitane is related to the genus [53,54] and confirms the chemosystematic possibility that this study has found an isomer of momordicoside L. The peak 22 showed a match in the GNPS library and it was annotated as hederagenin base-2H + 1O, O-AcetylHex. Peak 27 was suggested as a triterpene glycoside derivative, and we believe it has the same base structure of momordicoside L based on the molecular network. Furthermore, compound **27** has a difference of 4 Da concerning momordicoside L, suggesting two unsaturated bonds. Finally, the compound **23** [M − H]^−^ of *m/z* 721.4172 was annotated as aederagenin-O-acetyl-hex.

### 3.4. Bioactivity and Structure

Previous studies reported that flavonoids have antimicrobial activity [55,56]. These stand out even more because of their antibacterial properties, especially against strains of Gram-negative bacteria, which are responsible for serious opportunistic infections and are resistant to common therapies. In this sense, the study of plants with high flavonoid content should be highlighted [57]. Thus, our study focused on the ethyl acetate phase (MIC 156.2 µg·mL^−1^) against *K. pneumoniae*, which stands out for its ability to develop enzymatic resistance mechanisms and is considered to be largely responsible for several infectious diseases [58]. The PhEA proved to be rich in flavones, for instance quercetin-*O*-glucoside (**6**) and luteolin-O-rutinoside (**8**) (Figure 4), which confirms the correlation of the observed antimicrobial activity, as well the EE.

Our chemical prospecting data summarize chemical constituents that have in common a structural core of flavones, well known for a variety of activities [25]. Furthermore, isolated flavonols such as quercetin and kaempferol already show promising results as antimicrobials [59] as well against *K. pneumoniae* (MIC > 256 µg·mL^−1^) [60] and are the base structures of the main protagonists of the PhEA in this study. We emphasize carefully that the nominal results have a much more expressive value than many works given the same biological answer; perhaps the answer may be even more significant if the studies aim at obtaining isolated compounds. However, we emphasize that the rapid annotation provided using the LC-MS/MS technique does not require isolation, but shows an understanding of the active extension of the extract, directing more objective studies to that specific class. In addition, the literature [61] treats enriched phases as the main drivers for more specific studies, not to reach the main compounds responsible for the activity but to exclude those who may be acting as deterrents of the activity. In this sense, we are not being categorical in pointing only to these substances as active, but we are presenting a more interference-free sample.

## 4. Conclusions

This study showed the extract and phase ethyl acetate from *M. Charantia* leaves as an antibacterial agent. A total of 32 major compounds were detected, and, of these, 25 were annotated based on mass spectrometry data. Compounds including flavones, terpenes, organic acids, and inositol pyrophosphate derivatives are reported for the first time for the genus *Momordica*. The ethanolic extract exhibited low activity against *Proteus mirabilis* and *Klebsiella pneumoniae*. However, the phase ethyl acetate enriched with flavones showed interesting antibacterial inhibition against *K. pneumoniae*. Hence, we show that the leaves are a renewable antibacterial source and can serve as a pillar for future studies.

## Figures and Tables

**Figure 1 pharmaceutics-14-01796-f001:**
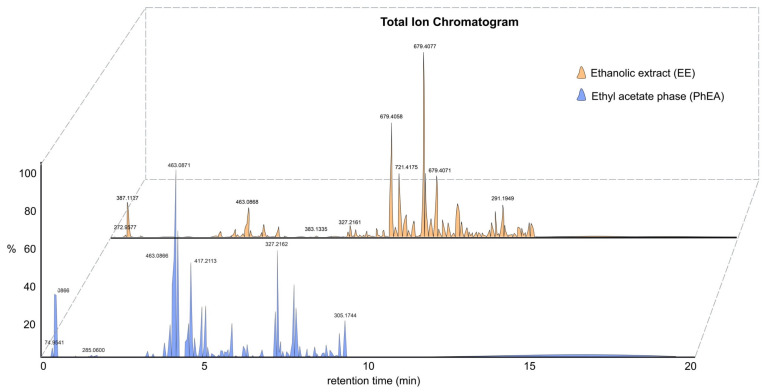
Total ion chromatogram (TIC) in the negative ionization mode of the ethanolic extract (EE), and ethyl acetate phase (PhEA).

**Figure 2 pharmaceutics-14-01796-f002:**
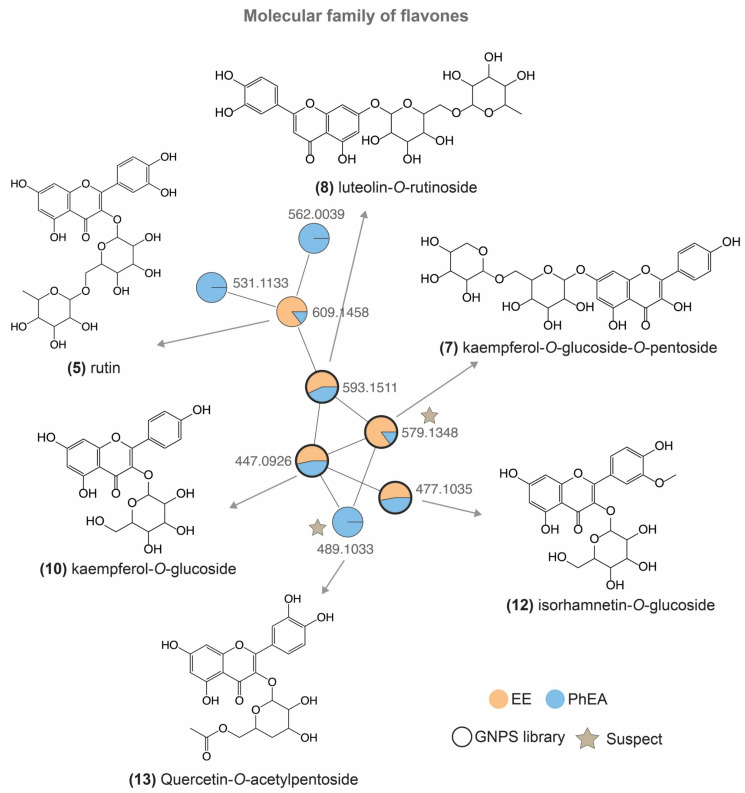
Feature-based molecular networking (GNPS) based on MS/MS data (ESI^−^) from GNPS library [21] and suspect list [38]. The arrows indicate the nodes that have a match in the spectral library and their respective structure. Node text indicates the parent ion, node color reports the extract of the plant (orange: ethanolic extract from leaves; blue: phase ethyl acetate from leaves).

**Figure 3 pharmaceutics-14-01796-f003:**
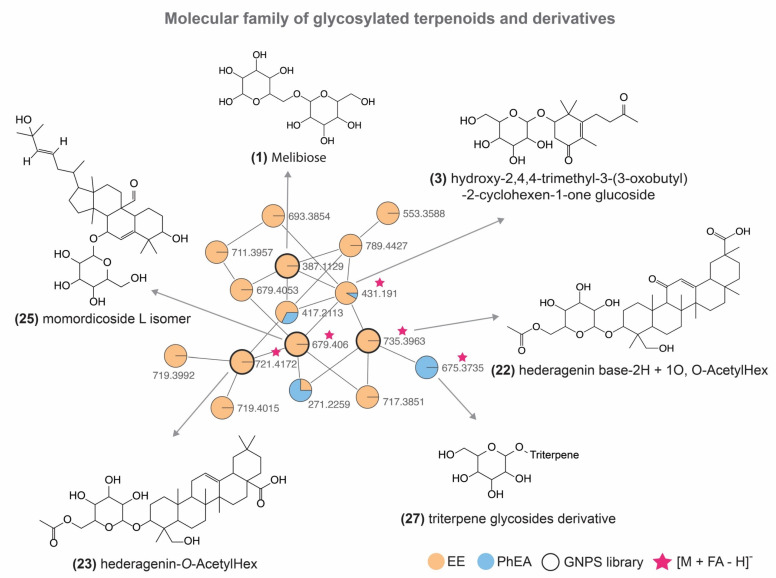
Representation of hydroxy-2,4,4-trimethyl-3-(3-oxobutyl)-2-cyclohexen-1-one glucoside (**3**) clusters in ethanolic extract using MS/MS in negative ionization mode.

**Figure 4 pharmaceutics-14-01796-f004:**
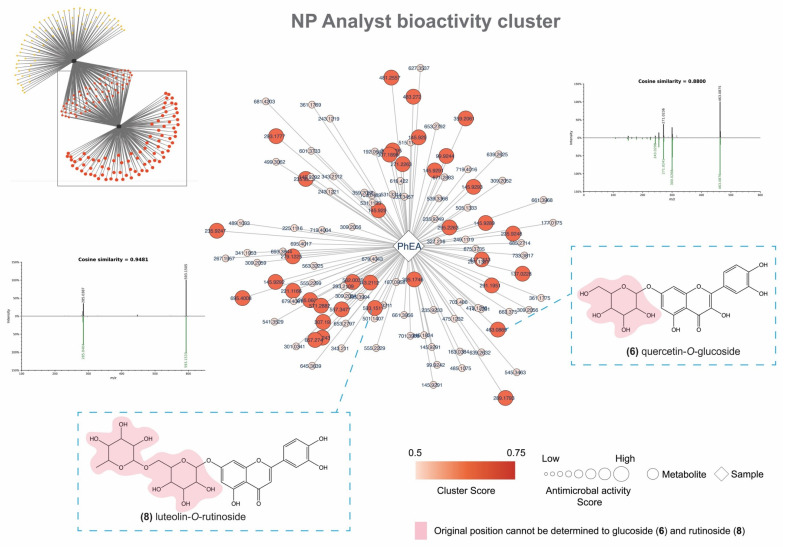
Community network. PhEA illustrated as square white node. Two examples of predicted bioactive flavones (**6**,**8**). The color of metabolite nodes is defined by cluster score (0.5–0.75). The size corresponds to the activity score (normalized data).

**Table 1 pharmaceutics-14-01796-t001:** Bacteria growth behavior in the presence of the extract and phases at different concentrations.

Concentration (µg·mL^−1^)	EE	PhHex	PhEA	PhWOH	EE	PhHex	PhEA	PhWOH	EE	PhHex	PhEA	PhWOH
	*Klebsiella pneumoniae*				*Proteus mirabilis*				*Staphylococcus aureus*			
2500	=	+	=	+	=	+	−	+	−	+	−	+
1250	=	+	=	+	=	+	+	+	−	+	+	+
625	=	+	=	+	=	+	+	+	−	+	+	+
312.5	+	+	=	+	=	+	+	+	+	+	+	+
156.2	+	+	=	+	+	+	+	+	+	+	+	+
78.1	+	+	+	+	+	+	+	+	+	+	+	+
39.0	+	+	+	+	+	+	+	+	+	+	+	+

Note: NP: natural product; EE: ethanolic extract; PhHex: hexane phase; PhEA: ethyl acetate phase; PhWOH: hydroalcoholic phase; = bactericidal effect; − bacteriostatic effect; + not active.

**Table 2 pharmaceutics-14-01796-t002:** The identified or tentatively identified compounds of ethanolic extract and ethyl acetate phase from *Momordica charantia* by LC-HRMS.

Peak	Rt (Min)	Molecular Formula	[M − H]^−^ (*m/z*)			Main Product Ions (MS/MS)	Annotated Compound	EE	PhEA
			Calculated	Accurate	Error (ppm)				
1	0.42	C_13_H_24_O_13_	387.1139	387.1129	2.6	341, 278, 179	melibiose	x	-
2	1.63	C_12_H_14_O_8_	285.0610	285.0602	2.8	153	dihydrobenzoic acid pentose	x	x
3	3.23	C_20_H_32_O_10_	^a^ 431.1917	^a^ 431.1910	1.6	385.1856 [M − H]^−^, 223, 205, 163, 119, 113	hydroxy-2,4,4-trimethyl-3-(3-oxobutyl)-2-cyclohexen-1-one glucoside	x	x
4	3.71	C_26_H_28_O_16_	595.1299	595.1306	1.2	445, 300, 272, 251, 191, 178	quercetin-*O*-sambubioside	x	x
5	3.97	C_27_H_30_O_16_	609.1456	609.1458	0.3	463, 301	rutin	x	x
6	4.08	C_21_H_20_O_12_	463.0877	463.0869	1.7	301, 271, 179	quercetin-*O*-glucoside	x	x
7	4.14	C_26_H_28_O_15_	579.1350	579.1348	0.3	463, 399, 327, 285, 151, 109	kaempferol-*O*-glucoside-*O*-pentoside	x	-
8	4.39	C_27_H_30_O_15_	593.1506	593.1511	0.8	547, 447, 357, 327, 285	luteolin-*O*-rutinoside	x	x
9	4.37	C_23_H_22_O_13_	505.0982	505.0980	0.4	300, 271, 255, 243, 178, 151	quercetin-*O*-glucosyl-6′′-acetate	x	x
10	4.53	C_21_H_20_O_11_	447.0927	447.0926	0.2	327, 284, 255, 227	kaempferol-*O*-glucoside	x	x
11	4.65	C_7_H_6_O_3_	137.0239	137.0228	8.0	93	4-hydroxybenzoic acid	x	-
12	4.67	C_22_H_22_O_12_	477.1033	477.1035	0.4	431, 357, 315, 300, 285, 271, 151	isorhamnetin-*O*-glucoside	x	-
13	4.87	C_23_H_22_O_12_	489.1038	489.1033	1.0	285, 255, 227	quercetin-*O*-acetylpentoside	x	x
14	4.98	C_20_H_34_O_9_	^a^ 417.2125	^a^ 417.2113	2.9	371.2052 [M − H]^−^, 209, 161,	icariside B6	x	x
15	5.66	C_18_H_32_O_7_	359.2070	359.2061	2.5	343, 305, 287, 239, 227, 209, 197, 171	unknown	x	-
16	5.77	C_15_H_10_O_7_	301.0348	301.0341	2.3	273, 245, 193, 179, 151, 121	quercetin	x	x
17	7.14	C_18_H_32_O_5_	327.2171	327.2162	2.7	291, 229, 171	trihydroxy octadecadienoic acid isomer	x	x
18	7.24	C_18_H_32_O_5_	327.2171	327.2160	3.3	291, 229, 171	trihydroxy octadecadienoic acid isomer	x	x
19	7.64	C_18_H_34_O_5_	329.2328	329.2320	2.4	211, 171	trihydroxy octadecenoic acid	x	x
20	8.38	C_37_H_60_O_11_	679.4057	^a^ 679.4053	0.6	633.3994 [M − H]^−^, 285	momordicoside L isomer	x	-
21	8.58	C_18_H_28_O_4_	307.1909	307.1903	2.0	289, 267, 235, 209, 185	unknown	x	-
22	9.06	C_39_H_60_O_13_	^a^ 735.3956	^a^ 735.3963	0.9	689.3918 [M − H]^−^, 667, 599, 527, 339	hederagenin base-2H + 1O, O-AcetylHex	x	-
23	9.74	C_39_H_62_O_12_	721.4163	721.4156	1.0	675 [M − H]^−^, 633, 513, 275, 193	hederagenin-*O*-AcetylHex	x	-
24	9.18	C_18_H_26_O_4_	305.1753	305.1746	2.3	287, 249, 207	unknown	x	-
25	9.37	C_37_H_60_O_11_	^a^ 679.4057	^a^ 679.4060	0.4	633.4015 [M − H]^−^, 575, 549, 471, 343	momordicoside L isomer	x	-
26	9.92	C_46_H_56_O_6_	703.3999	703.4061	0.6	659, 633, 597, 482, 350	unknown	x	-
27	10.0	C_30_H_60_O_16_	^a^ 675.3803	^a^ 675.3735	10	629.3677 [M − H]^−^, 569, 467, 447, 339, 297	triterpene glycosides derivative	x	x
28	10.4	C_37_H_60_O_11_	^a^ 679.4057	^a^ 679.4052	0.7	633.4084 [M − H]^−^, 530, 339, 291, 137	momordicoside L isomer	x	-
29	11.51	C_18_H_29_O_3_	293.2117	293.2109	2.7	275, 235, 183, 121	unknown	x	-
30	11.56	C_28_H_62_O_21_	733.3705	733.3729	3.3	689, 554, 412, 364, 259, 175	unknown	x	-
31	12.0	C_36_H_54_O_10_	645.3639	645.3639	0.0	601, 559, 513, 407, 339, 243, 168, 127	unknown	x	-
32	14.9	C_32_H_44_O_9_	571.2907	571.2882	4.4	525, 481, 391, 325, 315, 255, 241, 153	1-Hexadecanoyl-sn-glycero-3-phospho-(1′-myo-inositol) isomer	x	-

Note: ^a^ [M + HCOOH − H]^−^; EE: ethanolic extract; PhEA: phase ethyl acetate; x: presence; -: absent.

## Data Availability

All supporting data used in this study are available from the authors.

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
