# Peer review of "Antibacterial Activity from Momordica charantia L. Leaves and Flavones Enriched Phase"

_pharmaceutics, 2022, doi:10.3390/pharmaceutics14091796_

Round 1
Reviewer 1 Report
Review of the article: Antibacterial activity from Momordica charantia L. leaves and flavones enriched phase
Manuscript ID: pharmaceutics-1833071
In my opinion, the proposed manuscript is very interesting and generally well prepared. The “chemical part of the study” – extraction and determination of composition of produced extract was planned, performed and presented excellent. However, I have some questions about “microbial part” of the investigation. Below I have presented detailed comments.
Detailed comments
Abstract – In my opinion the authors should mention that produced extracts exhibited low activity (or it was not observed at all) for other investigated bacteria – e.g. S. aureus.
Introduction – well prepared no critical remarks.
Materials and methods
Lines 100-101 – was the produced mixture clear - did the authors observe precipitation of any compounds of the crude extract?
Lines 103 – 104 – how was the concentration of extract in these fraction determined?
Line 114 – it is not clear (in this place) if only crude extract was analysed.
Line 124 – the dot should be removed
Lines 157-158- what about human beings?
Line 162 – I would prefer titration plates rather than ELISA plates
Lines 171-175 – in my opinion concentration of DMSO in the first well (with the highest concentration of the extract/phase was a bit to high (5%)
Line 176- what was the concentration of the bacterial cells in the suspension (how was the suspension prepared)?
Line 184 – it was not “extract” – what volume was used for BHI agar plates inoculation
Line 193 – the full name of Proteus mirabilis was presented earlier (line 155), the same comment for S. aureus line 206
Results and discussion
Line 204 – what kind of leaves/seeds do the author mean? – the same plant? If yes how the authors explain that in previous study (citation 20) the highest activity was noticed for S. aureus?
In this part of the manuscript the authors should present explanation why so important differences were observed for selected species of bacteria.
Conclusions – similarly as in the case of introduction it should be mentioned tht some species of bacteria did not exhibit susceptibility to produced extracts.
Final decision – major revision
Author Response
Reviewer 1
In my opinion, the proposed manuscript is very interesting and generally well prepared. The “chemical part of the study” – extraction and determination of the composition of the produced extract was planned, performed, and excellently presented. However, I have some questions about the “microbial part” of the investigation. Below I have presented detailed comments.
Detailed comments
Reviewer 1: Abstract, in my opinion, the authors should mention that produced extracts exhibited low activity (or it was not observed at all) for other investigated bacteria – e.g. S. aureus.
Authors: Dear reviewer, a statement was written in the abstract
Reviewer 1, Lines 100-101: was the produced mixture clear - did the authors observe precipitation of any compounds of the crude extract?
Authors: Dear reviewer, the authors observed no precipitation of compounds during that process. Perhaps because that solution was sonicated using an ultrasonic bath to get a clear mixture. Moreover, our results confirmed flavones and triterpenes glycosylated, which in reason of so many hydroxyl groups are perfectly solubilized in H2O/Ethanol/HCl (% ~ 74.0:25.0:1.0).
Reviewer 1, Lines 103-104: how was the concentration of extract in this fraction determined?
Authors: Dear reviewer, we are not sure what you mean. However, all fraction concentrations were measured based on yield mass in mg from 1 g of extract. Moreover, the yielding mass was different for all fractions due to the partition coefficient. Please, let us know if we answered your question.
Reviewer 1, Line 114: it is not clear (in this place) if the only crude extract was analyzed.
Authors: Thank you for your kind comment. It was corrected in the main text to “The injection volume of the samples was 5 μL”
Reviewer 1, Line 124: the dot should be removed
Authors: It was corrected.
Reviewer 1, Lines 157-158: what about human beings?
Authors: It was corrected in the main text to “These microorganisms were selected for being common pathogens that can infect humans, animals or plants”
Reviewer 1, Line 162: I would prefer titration plates rather than ELISA plates.
Authors: Dear reviewer, using your suggestion, the authors charged from ELISA plates to 96-well microtitration plates.
Reviewer 1, Lines 171-175: in my opinion, the concentration of DMSO in the first well (with the highest concentration of the extract/phase was a bit too high (5%)
Authors: Dear reviewer, thanks a lot for your kind comment, we really appreciate that. However, we would say the concentration of DMSO which can affect bacterial growth depends on each species, as reported by Negin et al. (2015), different organisms respond differently to the same concentration of a specific solvent. Furthermore, Dong et al. (2013) highlighted that DMSO has lower toxicity than other solvents. In the literature, it is possible to find articles that used a higher volume of DMSO in the antimicrobial assay, for instance, Al-Bakri (2007) run his antimicrobial assay with 7.8% (v/v) of DMSO; ModrzyÅ„ski (2019) indicated that DMSO can be safely used as a co-solvent for short-term (≤3.5 h) bacterial growth inhibition assays at concentrations up to 3.3-10% (v/v). Additionally, in our laboratory, all the bacterias were tested with different concentrations of DMSO, to determine the concentration that does not affect their growth and interfere to the measurements. We highlighted that our study is agreed to the protocol published in Pharmaceutics | MDPI by Liu et al. (2020), hence, it is safe to say that the concentration of DMSO used in our study did not affect the antimicrobial assay.
References
Al-Bakri, A. G., & Afifi, F. U. (2007). Evaluation of the antimicrobial activity of selected plant extracts by rapid MTT colorimetry and bacterial enumeration. Journal of Microbiological Methods, 68(1), 19-25. https://doi.org/10.1016/j.mimet.2006.05.013
Dong Y, Wang J, Ding L, Liu Y (2013) Influence of Cosolvents on Low Water-Solubility Chemicals to Photobacterium phosphoreum in Acute Toxicity Test. Procedia Environ Sci 18:143–148. https://doi.org/10.1016/j.proenv.2013.04.019
Liu, J., Du, C., Beaman, H. T., & Monroe, M. B. B. (2020). Characterization of phenolic acid antimicrobial and antioxidant structure-property relationships. Pharmaceutics, 12(5), 419. https://doi.org/10.3390/pharmaceutics12050419
Modrzyński, J. J., Christensen, J. H., & Brandt, K. K. (2019). Evaluation of dimethyl sulfoxide (DMSO) as a co-solvent for toxicity testing of hydrophobic organic compounds. Ecotoxicology, 28(9), 1136-1141. https://doi.org/10.1007/s10646-019-02107-0
Negin, S., Gokel, M. R., Patel, M. B., Sedinkin, S. L., Osborn, D. C., & Gokel, G. W. (2015). The aqueous medium-dimethylsulfoxide conundrum in biological studies. RSC advances, 5(11), 8088-8093. https://doi.org/10.1039/C4RA15217D
Reviewer 1, Line 176: what was the concentration of the bacterial cells in the suspension (how was the suspension prepared)?
Authors: In each well, 5 μL of the bacterial suspension (104 Colony Forming Unit CFU/ml as required by CLSI) were inoculated and adjusted to the 0.5 McFarland Standard scale, and then, the plates were incubated at 37° C for 24 hours. That description was added in the main text.
Reviewer 1, Line 184: it was not “extract” – what volume was used for BHI agar plates inoculation
Authors: The type of activity presented in each concentration (bacteriostatic or bactericidal) was checked. In the cavities where there was no red color caused by the reaction of TTC with the bacteria, 5 μL of the wells that didn’t react with TTC were (re) inoculated in a Petri dish containing BHI agar culture medium and incubated at 37 °C for 24 h.
Reviewer 1, Line 193: the full name of Proteus mirabilis was presented earlier (line 155), the same comment for S. aureus line 206.
Authors: It was corrected.
Reviewer 1, Line 204: what kind of leaves/seeds does the author mean? – the same plant? If yes how do the authors explain that in the previous study (citation 20) the highest activity was noticed for S. aureus?
Authors: It corresponds to the seeds of the same plant. However, those authors used aqueous extracts prepared in phosphate buffer saline (PBS) and found peptides. Hence, this group of compounds has been reported for their activity against S. aureus (see Hou et al. 2007). Unlike our study, herein we are reporting glycosylated flavones and glycosylated triterpenes, a different group of molecules, as well their biological activities.
Reference
- Hou, S. Yonghui, P. Zhai, and G. Le. Inhibition of food-borne pathogens by Hf-1, a novel antibacterial peptide from the larvae of the housefly (Musca domestica) in medium and orange juice. Food Control, 18 (2007), pp. 1350-1357, https://doi.org/10.1016/j.foodcont.2006.03.007
Reviewer 1: The authors should present an explanation of why so important differences were observed for selected species of bacteria.
Authors: Dear reviewer, we really appreciate your suggestion. A statement was added in the 3.1. section. “We highlighted the bactericidal effect of Klebsiella Pneumonie, and Proteus Mirabilis, both Gram-negative strains. The literature (Tortora, 2018) reported that cell wall of Gram-negative bacteria acts as a barrier to a number of substances, including antibiotics. However, recently it was confirmed that quercetin derivatives have strong antibacterial action against Gram-negative bacteria (Osonga, 2019), and the ethyl acetate phase (PhEA, see table 1) described in this study showed high content of quercetin (16) derivatives, i.e. querce-tin-O-sambubioside (4), quercetin-O-glucoside (6), quercetin-O-glucosyl-6''-acetate (9), and quercetin-O-acetylpentoside (13)”.
References
Tortora, G.J.; Funke, B.R.; Case, C.L.; Weber, D.; Bair, W. Microbiology; 13th ed.; Pearson, 2018; ISBN 9780134605180.
Osonga, F.J.; Akgul, A.; Miller, R.M.; Eshun, G.B.; Yazgan, I.; Akgul, A.; Sadik, O.A. Antimicrobial Activity of a New Class of Phosphorylated and Modified Flavonoids. ACS Omega 2019, 4, 12865–12871, doi:10.1021/acsomega.9b00077.
Reviewer 1: Conclusions, similarly as in the case of the introduction it should be mentioned that some species of bacteria did not exhibit susceptibility to produced extracts.
Authors: It was added in the conclusion.
Reviewer 1: Final decision, major revision
Authors: Dear reviewer, we greatly appreciate your contributions to improving the presentation of our work. We hope that this time the writing is representative of our work. Besides, some details from the methods you have asked for were described. Thank you for your valuable contributions.

Reviewer 2 Report
Antibacterial activity from Momordica charantia L. leaves and flavones enriched phase
In this study, the authors demonstrated the evaluation of antibacterial activity and metabolites annotation for the ethanolic extract and ethyl acetate phase obtained from the leaves of Momordica charantia L. (Cucurbitaceae) against different pathogenic bacteria. The results of the present study indicate M. charantia leaves could be considered as an alternative antibacterial source against K. pneumoniae. I found that the paper has significant novelty and can be considered for publication in this journal. The manuscript is very well written, the authors have provided significant data to validate their hypothesis. However, I have few suggestions that authors may include during the revision.
1. The results part of the abstract needs to improve.
2. Paragraph 3 - Line number 52 to 68 – Author given details about chemical characterization. However, the author does not mention any reference, including suitable references.
3. Authors, please include some more Pharmaceutics activities of Momordica charantia L. leaves in the introduction part.
4. How does concentration of the Momordica charantia extract effect on tested pathogens?
5. Author, include some more discussion on antibacterial properties of Momordica charantia leaves and their phytocompounds.
6. Author, include the mechanisms of antimicrobial activity of the phase ethyl acetate enriched with flavones on tested pathogens.
7. Based on my knowledge, isolation of single compounds from plant extract is still expensive and it takes more time and energy. What is the author's opinion on this?
8. What is the biocompatibility of Momordica charantia extract and phytocompounds?
Author Response
Reviewer 2
In this study, the authors demonstrated the evaluation of antibacterial activity and metabolites annotation for the ethanolic extract and ethyl acetate phase obtained from the leaves of Momordica charantia L. (Cucurbitaceae) against different pathogenic bacteria. The results of the present study indicate M. charantia leaves could be considered as an alternative antibacterial source against K. pneumoniae. I found that the paper has significant novelty and can be considered for publication in this journal. The manuscript is very well written, and the authors have provided significant data to validate their hypothesis. However, I have a few suggestions that the authors may include during the revision.
Authors: Dear reviewer, thank you very much for your valuable contribution to the improvement of our manuscript. We carefully evaluate your recommendations and made the indicated corrections, which we highlighted in change control in the new version submitted for your conference. We also took the opportunity to re-write, review the English, and check other points that are included in the new version submitted.
Reviewer 2: The results part of the abstract needs to improve.
Authors: Dear reviewer, the abstract was rephrased as well improved according your suggestions. The authors are thankful to you.
Reviewer 2: Paragraph 3 - Line number 52 to 68 – The author has given details about chemical characterization. However, the author does not mention any references, including suitable references.
Authors: The suitable references were added.
Reviewer 2: Authors, please include some more Pharmaceutics activities of Momordica charantia L. leaves in the introduction part.
Authors: it was included in the introduction.
Reviewer 2: How does the concentration of the Momordica charantia extract affect tested pathogens?
Authors: Dear reviewer, we really appreciate your suggestion. We observed that Ethyl acetate phase enriched with glycosylated flavones has strong antibacterial properties, and it can be explained by quercetin derivatives which are able to break the cell wall of Gram-negative bacteria (Osonga, 2019). Moreover, as you can see in Table 1, concentrations below 625 µg.mL-1 of the Ethanolic Extract and below of 156.2 µg.mL-1 for the Ethyl acetate phase has bacteriostatic effects.
Reference
Osonga, F.J.; Akgul, A.; Miller, R.M.; Eshun, G.B.; Yazgan, I.; Akgul, A.; Sadik, O.A. Antimicrobial Activity of a New Class of Phosphorylated and Modified Flavonoids. ACS Omega 2019, 4, 12865–12871, doi:10.1021/acsomega.9b00077.
Reviewer 2: Author, include some more discussion on the antibacterial properties of Momordica charantia leaves and their phytocompounds.
Authors: it was included in the 3.1. section.
Reviewer 2: The author, includes the mechanisms of antimicrobial activity of the phase ethyl acetate enriched with flavones on tested pathogens.
Authors: Dear reviewer, we really appreciate your suggestion. However, we are talking about a phase of ethyl acetate flavones enriched, and it is not possible to discuss the mechanisms of antimicrobial activity, because it is generally an approach used to isolated compounds. Therefore, herein, our discussion was based on annotated flavones for the phase of ethyl acetate and their previous studies which reported flavonoids to have antimicrobial activity, especially against strains of gram-negative bacteria.
Reviewer 2: Based on my knowledge, isolation of single compounds from plant extract is still expensive and it takes more time and energy. What is the author's opinion on this?
Authors: Dear reviewer, our thoughts concerning how to discover a new bioactive product from plants changed due to the last pandemic, i.e. COVID-19 outbreak showed how science is fast-moving (Kupferschmidt, 2022). Therefore, the isolation of a single bioactive compound from the plant extract is still so expensive, mainly in our country, and it can take weeks, months, and years to disclose the compound’s identities as well as to get enough mass for biological assays. Thus, the authors have been minding to develop a workflow by LC-MS and advanced bioinformatic tools/approaches for the fast characterization of bioactive extract, fraction, and subfraction. Hence, we believe it can be the new state-of-the-art in the natural products field to disclose new bioactive products plant-based.
Kupferschmidt, Kai. n.d. “‘A Completely New Culture of Doing Research.’ Coronavirus Outbreak Changes How Scientists Communicate.” Accessed July 30, 2022. https://doi.org/10.1126/science.abb4761
Reviewer 2: What is the biocompatibility of Momordica charantia extract and phytocompounds?
Authors: Dear reviewer, we greatly appreciate your question. We would say that here, a group of phytocompounds (flavones derivatives) was related to a natural product (ethyl acetate phase) with high biocompatibility, as such as most natural products are very well accepted in clinical applications because its time-tested safety and efficacy (e.g. Aspirin, Morphine, Artemisinin, Dronabinol, Cannabidiol and so on). However, isolation of a single bioactive compound from the ethyl acetate phase is still so expensive and it takes weeks, months, and years to disclose its identity as well as to get enough mass for biological assays. Thus, the greatest interest of our group is to develop in the future steps of this project, a formulation containing the ethyl acetate phase of M. charantia, aiming to generate an herbal product with antibacterial action that can benefit society in general. Of course, many steps are still needed for that (in vivo assays, clinical assays, and so on), as well as grant support. Hence, we understand that the characterization of the extract and bioactive ethyl acetate phase is a great start, and it becomes sufficient, so far, for this purpose and bio-guide the future steps of this project.

Round 2
Reviewer 1 Report
The authors addressed all my comments and suggestions. The revised version of the manuscript is much better compared to the original text of the article. In my opinion the manuscript can be accepted in current form.